ecology/microbiology/plant science

assembly process, ectomycorrhiza, environmental gradient, fungal community, host preference, spatial structure

**Author for correspondence:**
Shunsuke Matsuoka
e-mail: code_matuoka@hotmail.com

# Evaluation of host effects on ectomycorrhizal fungal community compositions in a forested landscape in northern Japan

Shunsuke Matsuoka[1], Yoriko Sugiyama[2], Ryunosuke Tateno[3], Shihomi Imamura[4], Eri Kawaguchi[5] and Takashi Osono[6]

[1]Graduate School of Simulation Studies, University of Hyogo 7-1-28 Minatojima-minamimachi, Chuo-ku, Kobe 650-0047, Japan
[2]Graduate School of Human and Environmental Studies, Kyoto University, Kyoto 606-8501, Japan
[3]Field Science Education and Research Center, and [4]Graduate School of Agriculture, Kyoto University, Kyoto 606-8502, Japan
[5]Department of Life Science Frontiers, Center for iPS Cell Research and Application, Kyoto University, Kyoto 606-8507, Japan
[6]Department of Environmental Systems Science, Faculty of Science and Engineering, Doshisha University, Kyoto 610-0394, Japan

SM, 0000-0002-3723-8800

Community compositions of ectomycorrhizal (ECM) fungi are similar within the same host taxa. However, careful interpretation is required to determine whether the combination of ECM fungi and plants is explained by the host preference for ECM fungi, or by the influence of neighbouring heterospecific hosts. In the present study, we aimed to evaluate the effects of host species on the ECM community compositions in a forested landscape (approx. 10 km) where monodominant forest stands of six ECM host species belonging to three families were patchily distributed. A total of 180 ECM operational taxonomic units (OTUs) were detected with DNA metabarcoding. Quantitative multivariate analyses revealed that the ECM community compositions were primarily structured by host species and families, regardless of the soil environments and spatial arrangements of the sampling plots. In addition, 38 ECM OTUs were only detected from particular host tree species. Furthermore, the neighbouring plots harboured similar fungal compositions, although the host species were different. The relative effect of the spatial factors on the ECM compositions was weaker than that of host species. Our results suggest that

the host preference for ECM fungi is the primary determinant of ECM fungal compositions in the forested landscape.

## 1. Introduction

Ectomycorrhizal (ECM) fungi are symbionts of tree species belonging to the families Fagaceae, Betulaceae and Pinaceae, among others, and represent a dominant group of microorganisms inhabiting temperate and boreal forest floors [1]. ECM fungi play an essential role in plant growth and nutrient cycling by enhancing nutrient and water uptake from the soil to their host trees [2]. Since the function or ability of ECM fungi varies from species to species, the community responses of ECM fungi to environmental changes are critical for determining and maintaining forest ecosystem processes [3]. Various factors such as host taxa [4], soil properties (e.g. pH) [5] and dispersal limitation [6] have been proposed to affect the compositions of ECM fungal communities. For example, environmentally similar or spatially close sites are known to harbour similar ECM fungal communities [5–7]. Actually, the ECM fungal communities are simultaneously affected by each of these factors. Thus, researchers are now trying to quantify the effect of each factor on ECM fungal communities separately and have found significant effects of host trees on ECM communities [8–10].

The relationships between host tree species and ECM fungi have been repeatedly tested in a variety of regions and/or climatic zones [4,9–12]. Previous studies have investigated the relatedness of ECM fungal communities and host tree species, mainly in single forest stands (less than 1 ha) where several host species are mixed, by comparing associated ECM fungi among host individuals belonging to different taxa. These studies have shown that ECM community compositions are similar within the same host taxa [4,12]. For example, Tedersoo *et al.* [12] showed that host plants better explained variations in ECM fungal community composition than did soil environmental variables. Such compositional similarities in ECM fungal communities among the same host taxa have often been attributed to the preference of ECM fungi or host for particular partner species [13,14].

However, previous studies that investigated the effects of host species in a single mixed forest stand have not necessarily accurately evaluated the host's effects, owing to methodological limitations. First, the individuals of the same host species are likely to show clustered distribution in response to the local environmental conditions and past dispersion [15]. In these cases, the environmentally similar or spatially close sites tend to harbour similar host communities (i.e. the host community shows correlation with other factors), making it difficult to separate the effects of the host from those of other factors. Second, in mixed forests ECM fungal communities are inevitably affected by the surrounding host species. That is, the same ECM fungal individuals can be shared between adjacent trees via below-ground mycelia [16]. Furthermore, since most fungal spores fall within several metres from sporocarps [17], the spatially closer trees potentially share more inoculums. Therefore, ECM fungal compositions can be similar among spatially close host trees, regardless of the host taxa [18]. Thus, in most field studies, the effect of each factor has not been fully separated and the effect of the host has not been accurately evaluated [8], even though the effects of each factor on ECM fungal communities were evaluated simultaneously.

Among these problems, the correlation between the host and other factors, and the effects of surrounding host species, can be eliminated by conducting surveys in several patchily distributed monodominant forest stands. If the host species has a strong influence, the ECM composition will cluster by host species, regardless of the spatial arrangements of the forest stand. On the contrary, if other environmental factors or spatial distance have effects stronger than those of the host species, the ECM fungal community compositions should resemble those in the environmentally similar or spatially closer sites, regardless of the host species.

In the present study, we aimed to evaluate the effect of host trees on ECM fungal community compositions relative to soil environments and spatial distances in a forested landscape (approx. 10 km). Our study forests included monodominant stands of six ECM host species, including three broad-leaved tree species belonging to the families Fagaceae and Betulaceae, as well as three coniferous species belonging to the family Pinaceae. These forest stands are patchily distributed over the forest. In this setting, we analysed the following factors: (i) the effects of the host tree species belonging to three families on the community compositions of ECM fungi, (ii) the explanatory power of host tree identities on the ECM compositions relative to other environmental and spatial variables, and (iii) the relationships between individual ECM fungal species and host tree species.

# 2. Material and methods

## 2.1. Study sites and sampling procedure

The study was conducted in the Shibecha branch of Hokkaido Forest Research Station, Field Science Education and Research Center, Kyoto University, located in the eastern part of Hokkaido Island in northern Japan (1446.8 ha, 43°22′ N, 144°37′ E, approx. 25–150 m.a.s.l.). The forest area of the station extends approximately 9 km from south to north and approximately 1–3 km from east to west and is surrounded by a pasture. The 30-year mean annual temperature is 6.3°C, and the 30-year mean annual precipitation in the forest is 1188.7 mm. The 30-year mean monthly temperature is the highest in August (19.8°C) and the lowest in January (−9.0°C), and the 30-year mean monthly precipitation is the highest in September (181.8 mm) and the lowest in January (30.7 mm) (1986–2015, 43°19′ N, 144°36′ E, Kyoto University Forests 2016). Soils in the study sites are Andosols [19], and the soil texture is characterized as clay loam or loam.

The vegetation in natural, old growth forest is mainly composed of the following deciduous broad-leaved tree species: *Quercus crispula* Blume, *Ulmus davidiana* Planchon var. *japonica* (Rehder) Nakai, *Fraxinus mandschurica* Rupr. var. *japonica* Maxim. and *Acer pictum* Thunb. subsp. *dissectum* (Wesm.) H. Ohashi, as well as the following pioneer species: *Betula platyphylla* Sukaczev and *Alnus hirsuta* (Spach) Turcz. ex Rupr. These tree species are patchily distributed on clear-cut areas such as roadsides and timber yards. *Alnus* species are known to associate with a distinctive community of ECM fungi (e.g. [9]). The coniferous plantations are monocultures containing species such as *Larix kaempferi* (Lamb.) Carr., *Abies sachalinensis* F. Schmidt and *Picea glehnii* F. Schmidt that were planted from the 1960s to the 1980s in this forest station. *Abies sachalinensis* and *P. glehnii* are common species on Hokkaido Island, but are not naturally distributed in the forest station. *Larix kaempferi* does not occur naturally on Hokkaido Island, but was introduced from Honshu Island in Japan for afforestation. Approximately 70% of the total area of the forest station is covered by deciduous broad-leaved forests, and the remaining area is occupied by plantation forests in which tree species *L. kaempferi*, *A. sachalinensis* and *P. glehnii* cover approximately 14%, 8% and 2%, respectively.

Six tree species (three broad-leaved and three coniferous species) were targeted as host species. For each host species, three stands (approximately 0.4 ha), where the targeted species dominated as an ECM host species, were chosen as sampling plots (table 1 and figure 1). The latitude (43.3364–43.4061 N), longitude (144.6255–144.6627 E) and altitude (44.92–153.39 m) of each plot as well as the diameter at breast height (DBH) of individual trees were recorded. At each plot, we selected 10 host species individuals that had a DBH greater than 20 cm and collected a block of surface soil (10 × 10 × 5 cm from a depth of 5–10 cm), including tree roots, within 1 m from each tree trunk. All host tree individuals were spaced at least 3 m apart from each other, thus minimizing the spatial autocorrelation effect of individual ECM fungi [20]. The blocks were kept in plastic bags and frozen at −20°C during transport to the laboratory. A total of 180 blocks (6 host species × 3 study plots × 10 soil blocks) were used for the study.

In the laboratory, fine roots of trees were extracted from the soil samples using a 2 mm mesh sieve by gently washing with tap water to remove the soil particles and debris. In each block, 20 individual root segments (approximately 5 cm in length) were selected, and one ECM root tip (1 to 2 mm in length) was collected from each root segment under a 20× binocular microscope. The 20 ECM root tips obtained from each block were pooled and kept in a tube containing 70% ethanol (w/v) at −20°C. Before extracting DNA, the root tips were washed with 0.005% aerosol OT (di-2-ethylhexyl sodium sulfosuccinate) solution (w/v) and rinsed with sterile distilled water to remove small soil particles on the root surface. The root tips were then transferred to tubes containing cetyltrimethylammonium bromide (CTAB) lysis buffer and stored at −20°C until DNA extraction.

## 2.2. Soil properties

Mineral soils (0–10 cm in depth) were collected by a soil core sampler (surface area was 20 cm$^2$). Five soil core samples were collected at the distance interval of 1.5 m along a straight line from each plot and composited for each plot. The composited soil samples were dried at 70°C for more than 72 h and passed through a 4 mm mesh sieve to remove fine roots, pieces of organic matter and gravel. Total soil N and C were determined by an NC analyser (Sumigraph NC-900, Sumika Chemical Analysis Service, Ltd, Osaka, Japan), and the soil pH was determined by a pH meter (HORIBA D-51, Horiba,

**Table 1.** Host tree species and their stand conditions.

| plot | host tree species | host Family | soil pH | soil C/N ratio | elevation (m) | latitude (°N) | longitude (°E) | dominance of the host species[a] (%) | other ECM tree species[b] |
|------|-------------------|-------------|---------|----------------|---------------|---------------|----------------|-------------------------------------|---------------------------|
| Quercus 1 | Quercus crispula | Fagaceae | 5.40 | 13.6 | 118.49 | 43.3462 | 144.6526 | 58.0 | Salix caprea (3.3) Betula platyphylla (1.9) |
| Quercus 2 | Quercus crispula | Fagaceae | 5.22 | 13.3 | 121.13 | 43.3897 | 144.6552 | 44.5 | — |
| Quercus 3 | Quercus crispula | Fagaceae | 4.75 | 16.3 | 139.50 | 43.4026 | 144.6433 | 86.0 | — |
| Betula 1 | Betula platyphylla | Betulaceae | 5.32 | 12.5 | 44.92 | 43.3364 | 144.6255 | 57.0 | Alnus hirsute (18.7) |
| Betula 2 | Betula platyphylla | Betulaceae | 4.98 | 12.3 | 55.96 | 43.3573 | 144.6414 | 67.7 | Alnus hirsute (16.6) |
| Betula 3 | Betula platyphylla | Betulaceae | 5.47 | 12.8 | 72.91 | 43.3818 | 144.6454 | 97.7 | — |
| Alnus 1 | Alnus hirsuta | Betulaceae | 5.25 | 14.1 | 46.83 | 43.3375 | 144.6300 | 100 | — |
| Alnus 2 | Alnus hirsuta | Betulaceae | 5.38 | 13.2 | 65.79 | 43.3764 | 144.6462 | 100 | — |
| Alnus 3 | Alnus hirsuta | Betulaceae | 5.40 | 13.2 | 105.54 | 43.3984 | 144.6442 | 100 | — |
| Abies 1 | Abies sachalinensis | Pinaceae | 5.48 | 11.8 | 51.38 | 43.3389 | 144.6325 | 89.0 | — |
| Abies 2 | Abies sachalinensis | Pinaceae | 5.34 | 13.1 | 139.79 | 43.3698 | 144.6627 | 100 | — |
| Abies 3 | Abies sachalinensis | Pinaceae | 5.14 | 16.0 | 153.39 | 43.4061 | 144.6428 | 89.3 | Quercus dentate (3.7) |
| Picea 1 | Picea glehnii | Pinaceae | 5.28 | 15.3 | 51.65 | 43.3412 | 144.6325 | 96.0 | — |
| Picea 2 | Picea glehnii | Pinaceae | 5.05 | 13.8 | 116.71 | 43.3670 | 144.6508 | 98.7 | — |
| Picea 3 | Picea glehnii | Pinaceae | 5.17 | 13.2 | 133.28 | 43.3682 | 144.6514 | 98.9 | — |
| Larix 1 | Larix kaempferi | Pinaceae | 5.16 | 13.3 | 80.43 | 43.3426 | 144.6358 | 92.1 | Betula platyphylla (2.0) Salix caprea (0.8) |
| Larix 2 | Larix kaempferi | Pinaceae | 4.92 | 13.0 | 55.23 | 43.3570 | 144.6403 | 93.3 | — |
| Larix 3 | Larix kaempferi | Pinaceae | 4.54 | 13.5 | 134.69 | 43.3682 | 144.6625 | 99.1 | Betula platyphylla (0.9) |

[a]Calculated based on basal area (m$^2$ per ha).
[b]Dominance of each species based on basal area (m$^2$ per ha) are in parentheses.

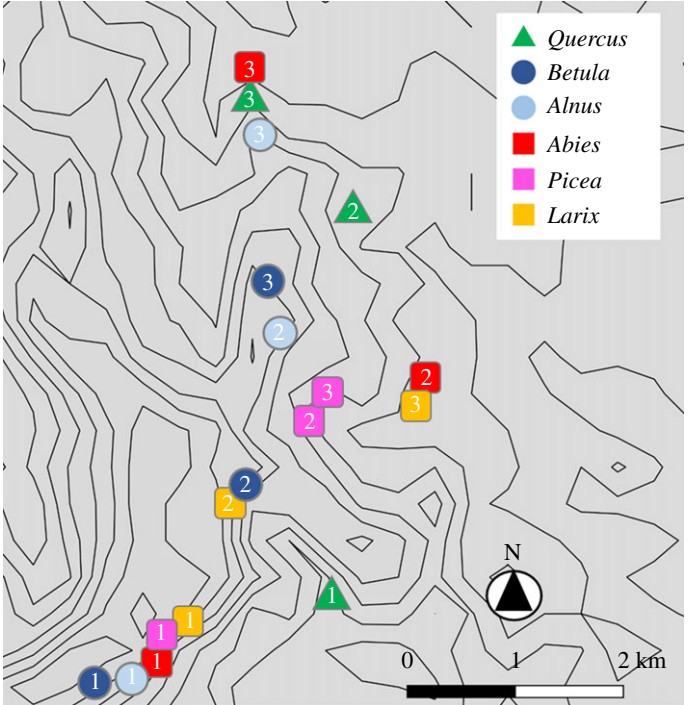

**Figure 1.** Sampling plots of each tree species. Plot numbers in the symbols are consistent with those listed in table 1 and figure 2.

Ltd, Kyoto, Japan) after extraction with deionized water at a dry soil : water ratio of 2 : 5 (w/w). Ranges of soil pH and C/N ratio of sampling plots are 4.54–5.48 and 11.8–16.3, respectively (table 1).

## 2.3. DNA extraction, PCR amplification and pyrosequencing

DNA analysis was generally performed according to methods described by Matsuoka *et al.* [8]. Whole DNA was extracted from root tips in 180 samples using the modified CTAB method described by Gardes & Bruns [21]. For the direct 454 pyrosequencing of the fungal internal transcribed spacer 1 (ITS1) [22], we used a semi-nested PCR protocol. First, the entire ITS region and the 5′-end region of the large subunit were amplified using the fungus-specific primers ITS1F [21] and LR3 [23]. PCR was performed in a 20 μL volume containing 1.6 μl of template DNA, 0.3 μl of KOD FX NEO (TOYOBO, Osaka, Japan), 9.0 μl of 2X buffer, 4.0 μl of dNTPs, 0.5 μl each of the two primers (10 μM) and 4.1 μl of distilled water. The PCR amplification was performed using the following conditions: an initial denaturation step at 94°C for 5 min, followed by 23 cycles of denaturation at 95°C for 30 s, annealing at 58°C for 30 s and extension at 72°C for 90 s, and then a final extension step at 72°C for 10 min. The PCR products were purified using the ExoSAP-IT PCR Product Clean-up Kit (GE Healthcare, Little Chalfont, Buckinghamshire, UK). Thereafter, the second PCR targeting the ITS1 region was performed using the ITS1F primer fused with an 8 bp DNA tag [24] and the universal primer ITS2 [25]. The second PCR was performed in a 20 μL volume containing 1.0 μl of template DNA, 0.2 μl of KOD Plus NEO (TOYOBO), 2.0 μl of 10X buffer, 2.0 μl of dNTPs, 0.8 μl each of the two primers (5 μM) and 13.2 μl of distilled water. The PCR conditions were as follows: an initial denaturation step at 94°C for 5 min, followed by 28 cycles of denaturation at 95°C for 30 s, annealing at 60°C for 30 s, extension at 72°C for 90 s, and a final extension step at 72°C for 10 min. The PCR products were pooled into five libraries and purified using an AMPure Magnetic Bead Kit (Beckman Coulter, California, USA). The pooled products were sequenced in two 1/8 regions using the GS-FLX sequencer (Roche 454 Titanium) at the Graduate School of Science, Kyoto University, Japan. The sequence data were deposited in the Sequence Read Archive of the DNA Data Bank of Japan (accession number: DRA007781).

## 2.4. Bioinformatics

The bioinformatics analyses were performed using the methods described by Matsuoka *et al.* [8]. Using the 454 pyrosequencing method, 272 358 reads were obtained. These reads were trimmed with a

minimum quality value of 27 at the 3′ tails [26] and sorted into individual samples using the sample-specific tags. The pyrosequencing reads were assembled using Claident pipeline v. 0.2.2018.05.29 [27]. First, the short reads (less than 150 bp), and then the potentially chimeric sequences and pyrosequencing errors were removed, using the software programs UCHIME [28] and CD-HIT-OTU [29], respectively. The remaining 204 627 reads were assembled at a threshold similarity of 97%, which is widely used for the fungal ITS region [30], and the resulting consensus sequences represented molecular operational taxonomic units (OTUs). Then, singleton OTUs were removed. The consensus sequences of the OTUs are listed in electronic supplementary material, table S1 (Supporting Information).

To systematically annotate the taxonomy of the OTUs, we used Claident v. 0.2.2018.05.29 [31], which was built upon an automated, basic local alignment search tool (BLAST) search using the National Center for Biotechnology Information (NCBI) BLAST+ algorithm [32] and a taxonomy-based sequence identification engine. Using the reference database from the International Nucleotide Sequence Database Collaboration (INSDC) for taxonomic assignment, the sequences homologous to the ITS sequence of each query were fetched, and then the taxonomic assignment was performed based on the lowest common ancestor algorithm [33]. The results of Claident and the number of reads for the OTUs identified are given in electronic supplementary material, table S1. To screen for ECM fungi, we referred to the reviews by Tedersoo et al. [34] and Tedersoo & Smith [35] and assigned OTUs to the genera and/or families that were predominantly ECM fungi. The resultant ECM fungal OTUs (ECM OTUs) were used for further analyses (see electronic supplementary material, table S1).

## 2.5. Data analyses

For all data analyses, the presence or absence of the ECM OTUs was used as the binary data, rather than the quantitative use of numbers generated from amplicon sequencing [36,37]. All analyses were performed using the R v. 3.4.4 [38]. Differences in the sequencing depth of individual samples affect the number of OTUs retrieved, often leading to the underestimation of OTU richness in the samples that had low sequence reads. In our dataset, because the rarefaction curves for all samples reached an asymptote (electronic supplementary material, figure S1), we did not conduct rarefaction analysis.

The OTU compositions were compared between plots. First, the presence or absence of ECM OTUs was recorded for each sample. Subsequently, these data for the presence or absence were merged within the plots, and the incidence data of each OTU for each plot were generated ($n = 10$ for each plot). The maximum occurrence of each OTU was 10 for a single plot. To examine the ECM OTU composition, the dissimilarity index of OTU composition between plots was calculated using the Bray–Curtis index in which the incidence of OTUs is considered. In addition, we used the Raup–Crick index in which only the presence or absence of individual OTUs at each plot was used to confirm the robustness of the results, regardless of the other dissimilarity indices used. The Raup–Crick dissimilarity index is a probabilistic index and is less affected by the species richness gradient among sampling units than are other major dissimilarity indices, including the Bray–Curtis index [39]. The community dissimilarity of ECM OTUs among plots was ordinated in non-metric multidimensional scaling (NMDS). The correlation of the NMDS structure with host identity and geographic (i.e. latitude and longitude) and environmental (i.e. elevation, soil pH and soil C/N ratio) variables was tested by permutation tests ('envfit' command in the vegan package, 9999 permutations). Subsequently, to investigate whether the dissimilarity of OTU composition is related to the host (species or family) and geographic positions of the plots (latitude and/or longitude), one-way permutational multivariate analysis of variance (PERMANOVA) was conducted.

We used variation partitioning based on the distance-based redundancy analysis (db-RDA, 'capscale' command in the vegan package) to quantify the contribution of the host, environmental and spatial variables to the community structure of ECM fungal OTUs. The relative weight of each fraction (pure, shared and unexplained fractions) was estimated following the methodology described by Peres-Neto et al. [40]. For the distance-based redundancy analysis (db-RDA), we constructed two models including environmental and spatial variables. The detailed methods for variation partitioning are described by Matsuoka et al. [8]. First, we constructed environmental models by applying the forward selection procedure (999 permutations with an alpha criterion = 0.05) of Blanchet et al. [41]. The full models were as follows: [pH + C/N ratio + elevation + host identity]. Thereafter, we constructed the models using spatial variables, which were extracted based on Moran's eigenvector maps (MEM) [42]. The MEM analysis produced a set of orthogonal variables derived from the geographical coordinates of the sampling locations. The MEM vectors were calculated using the 'dbmem' command in the adespatial package. We used the MEM vectors that best accounted for autocorrelation and then conducted forward selection (999 permutations with an alpha criterion = 0.05; the full model contained six MEM variables).

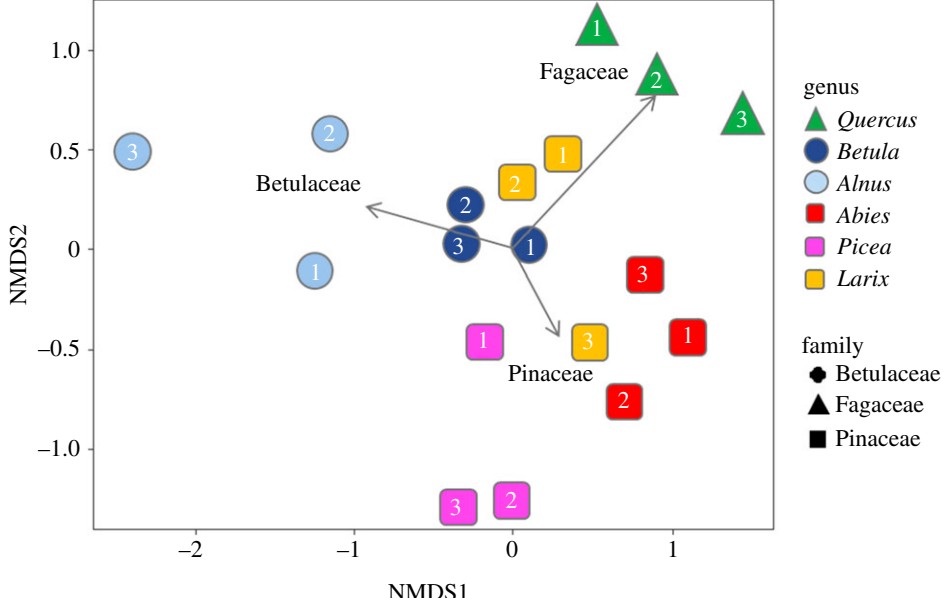

**Figure 2.** Community dissimilarity among the plots as revealed by NMDS ordination using the Bray–Curtis index (stress value = 0.125). Plot numbers in the symbols are consistent with those listed in table 1 and figure 1.

Based on these two models, we performed variation partitioning by calculating the adjusted $R^2$ values for each fraction [40].

To determine which OTU had significantly different frequencies among the host species, an indicator taxa analysis [43] was performed using the 'signassoc' function in the 'indicspecies' package on the presence or absence data for each sample ($n = 180$). We used mode = 1 (group-based) and calculated the $p$-values with 999 permutations after applying Sidak's correction for multiple testing.

# 3. Results

## 3.1. Taxonomic assignment

In total, the filtered 204 627 pyrosequencing reads from the 180 samples were grouped into 488 OTUs with 97% sequence similarity (electronic supplementary material, table S1). Among them, 180 OTUs (53 939 reads) belonged to the ECM fungal taxa, with 169 OTUs belonging to Basidiomycota, and 11 OTUs to Ascomycota. Each plot yielded 4 to 35 OTUs (18 OTUs on average). At the family level, 169 OTUs belonged to 20 families, with the common families being Thelephoraceae (75 OTUs, 41.7% of the total number of ECM fungal OTUs) and Russulaceae (26 OTUs, 14.4%). These two families accounted for 41.0–69.8% of the total richness of ECM fungal OTUs in each tree species (electronic supplementary material, figure S4).

## 3.2. Community structures of ECM OTUs

The NMDS ordination showed the separation of ECM OTU composition among plots (figure 2, stress value = 0.125). The ordination was significantly correlated with the host species and family ('envfit' function; host species, $R^2 = 0.851$, $p < 0.001$; host family, $R^2 = 0.559$, $p < 0.001$), but not with the latitude, longitude, elevation, soil pH and C/N ratio of the plot (latitude, $R^2 = 0.029$, $p = 0.795$; longitude, $R^2 = 0.052$, $p = 0.670$; elevation, $R^2 = 0.1456$, $p = 0.308$; soil pH, $R^2 = 0.1099$, $p = 0.4047$; soil C/N ratio, $R^2 = 0.0243$, $p = 0.8334$). In the PERMANOVA, both host species and host family significantly affected the ECM composition (host species, $F$-value = 57.7, $R^2 = 0.960$, $p < 0.001$; host family, $F$-value = 7.02, $R^2 = 0.484$, $p < 0.001$). In the variation partitioning, only host tree species identity was selected as an environmental variable, and two MEM vectors (MEM 4 and MEM 2) were selected as spatial variables. The percentages explained by the host tree species and spatial fractions were 28.7% and 5.4%, respectively, and no shared fraction was detected between the host tree species and spatial variables. In total, 34.1% of the community variation was explained and the remaining 65.9% was unexplained. The use of the Raup–Crick index did not affect

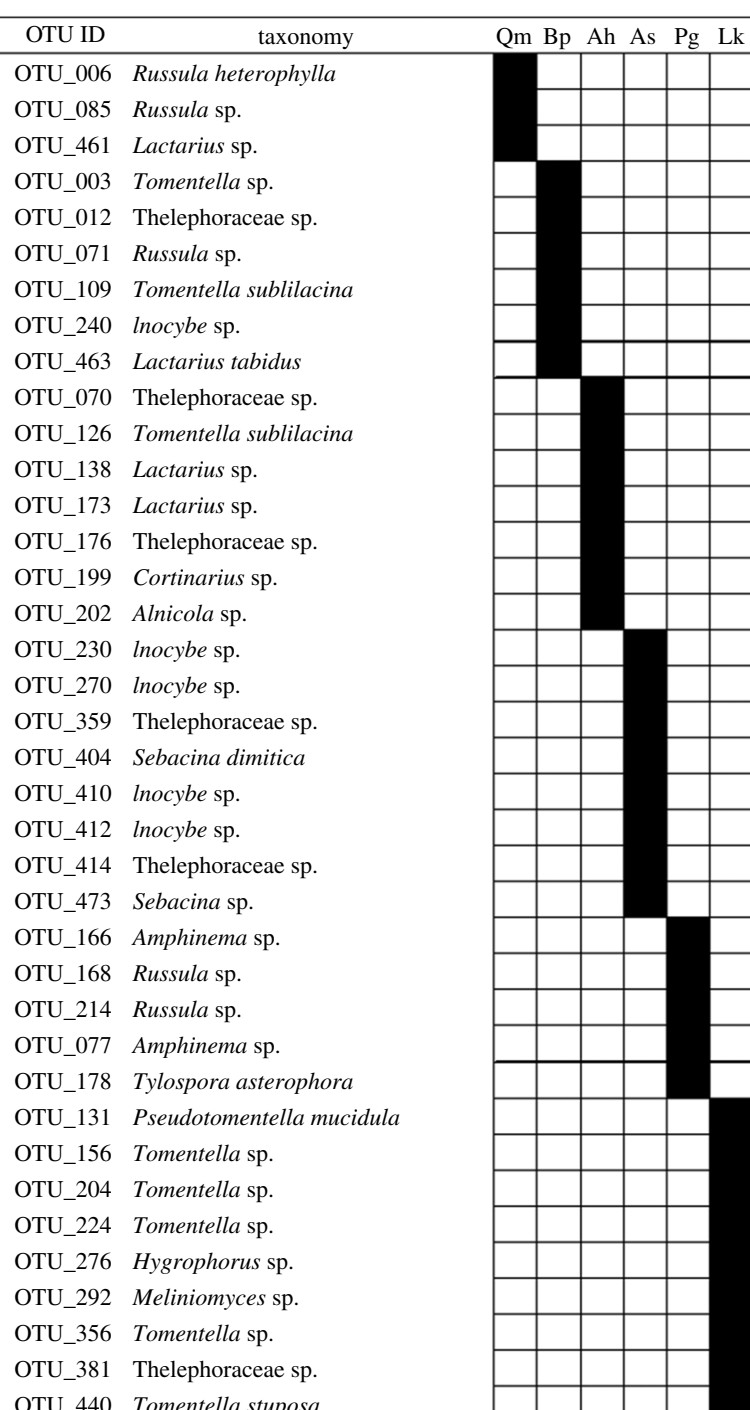

**Figure 3.** OTUs with significantly high detection frequency in particular host tree species. Filled boxes show the combination of ectomycorrhizal (ECM) OTU and tree species with significantly high detection frequency. ($p < 0.05$ after Sidak's correction). OTU ID and taxonomy are in accordance with electronic supplementary material, table S1. Qm *Quercus*, Bp *Betula*, Ah *Aluns*, As *Abies*, Pg *Picea*, Lk *Larix*.

the results. The NMDS ordination and results of variation partitioning with the Raup–Crick index are available in the electronic supplementary materials (electronic supplementary material, figures S2 and S3).

The indicator taxa analysis comparing the ECM communities among the host tree species detected significantly different host preferences of 38 OTUs ($p < 0.05$ after Sidak's correction, figure 3). For each tree species, three to nine ECM OTUs showed significantly higher frequencies of occurrence than the other tree species. Different ECM OTUs belonging to the same genus preferred different host tree species. (e.g. OTU_085, OTU_071 and OTU_168 belonging to the genus *Russula* preferred *Quercus*, *Betula* and *Picea* tree species as host trees, respectively). In addition, for OTU_109 and OTU_126

belonging to the same ECM fungal species, *Tomentella sublilacina*, the frequently detected host tree species were different, being *Betula* and *Alnus*, respectively.

# 4. Discussion

In the present study, we clearly showed the relationships between host species and the ECM fungal community composition by investigating the monodominant forest stands of six ECM host species. We quantitatively evaluated the effect of host tree species, abiotic environments and spatial factors on ECM fungal communities in the field, thereby demonstrating the relative importance of the host. In our study, the ECM fungal community composition was primarily explained by host species and/ or family. In variation partitioning, any part of the fraction explained by the host was not shared by the environmental or spatial factors, indicating that we successfully evaluated the pure effect of the host species in the present study. From this variation partitioning, we could infer the relatively higher importance of the host relative to other environmental and spatial factors. In addition, some OTUs were preferentially associated with specific host tree species in our field and could partly contribute to the compositional similarity of ECM fungi within the same host species.

Our results clearly demonstrated that the ECM fungal compositions were primarily determined by host species rather than the soil environments and spatial arrangements of the plots. Similar ECM fungal community composition within the same host species and/or phylogeny has also been detected in other sites and host taxa [4,8–10,12]. These similarities in ECM fungal community compositions have been related to the preference of the fungi and/or host tree to partner species; however, the exact mechanism of the preference has not been fully revealed, and we could not distinguish between fungal preference and plant preference in the present study. For example, Bogar *et al.* [13] conducted pot experiments with varying symbiotic ability among ECM fungal species and suggested that host plants can discriminate among fungal partners and reward more carbohydrates to the fungal species that are beneficial to them. Such selection of a fungal partner by host plants might lead to the different ECM fungal compositions among host species in the field. In addition to these direct interactions between the host tree and ECM fungus, environments that the host tree generates (e.g. soil properties) [44,45], or the interaction with other organisms under particular host species such as soil bacteria or fungus, might generate different ECM compositions among host tree species [46].

In our study, the host species has a primary effect on the ECM fungal community composition. However, as the present study is based on field observations, we cannot infer a causal relationship between a host species and an ECM fungal community. Especially, there is a possibility that unmeasured factors are related to ECM fungal community composition. For example, in the present study, because the stands of broad-leaved trees are natural forests, the differences in fungi among these stands might be because the fungi and tree species are independently adapted to the same environment [47]. Furthermore, in variation partitioning, 65.9% of the community difference remained unexplained. This unexplained fraction might include the effects of the vegetation in the surrounding area, unexplained environmental factors (e.g. soil organic phosphorus) [48] and drift (i.e. random arrival and extinction) [49]. In the present study, the effects of abiotic environments, such as soil, on ECM composition were relatively weak. This may be partly due to the relatively narrow range of environments in the study area [50] and an inadequate sampling effort in each plot.

In our site, the detection of some OTUs was biased to specific host species (figure 3). These OTUs might have a high host preference (figure 3). Different OTUs belonging to the same genus preferred different host tree species. For example, OTU_085, OUT_071 and OUT_168 belonging to the genus *Russula* preferred *Quercus*, *Betula*, and *Picea* tree species as host trees. Moreover, although OTU_109 and OTU_126 were identified as the same species (*Tomentella sublilacina*), the host species frequently associated with these two OTUs were different (*Betula* and *Alnus*, respectively). *Tomentella sublilacina* has been detected in association with various regions and host tree species in the Northern Hemisphere [11,47], and its preferred host tree species might be different among varying genotypes and/or habitats. Our results indicate that the degree of preference and the preferred host is different at the fungal species or genotype level, rather than at the genus or family level in our study forests.

Besides host species, the effect of spatial distance on the ECM fungal community composition was detected. This indicates that ECM fungal compositions become similar at spatially close sites, regardless of the host trees. For example, in the present study, the ECM fungal communities were similar between the *Betula* and *Larix* forests and between the *Abies* and *Larix* forests (figure 2). These high similarities in ECM compositions can be partly due to the geographical closeness of the *Betula* 2 and *Larix* 2 plots, and

between the *Abies* 2 and *Larix* 3 plots (*ca* 100 m, table 1 and figure 1). Dispersal and colonization limitations can be suggested as factors that lead to such spatial structures at a small spatial scale (*ca* less than 100 m). Although the dispersal distances of fungal spores are not fully understood, a previous study revealed that most spores fall within several metres from sporocarps [17]. Thus, spatially closer plots potentially share more inoculums. Moreover, in spatially closer plots, the same ECM fungal individuals can be shared between different host species via below-ground mycelia. Such sharing of inoculum and/or mycelia might result in the sharing of ECM species between different adjacent tree species [16,18]. In our study site, for example, OTU_109 was detected both from *Betula* 2 and *Larix* 2. This OTU is preferentially associated with *Betula* (figure 3); therefore, the detection of this OTU from the *Larix* plot might be due to the colonization via mycelia. As few studies have investigated the distance limitation in such colonization via mycelia, elucidating the importance and frequency of these colonizations to a non-preferred host needs further investigation. Nevertheless, in our results, the ECM fungal communities were shown to be similar between neighbouring plots situated within a distance of 100 m, although the host tree species were different. Such spatial structure can hinder the investigation of fungi–plant combinations caused by partner preferences.

In summary, in the present study, we clearly demonstrated that the ECM fungal communities were structured by the species and families of host trees in a forested landscape. Our results further suggest that the preference of fungi and/or host to partner species can structure ECM fungal compositions in the field. Additionally, in our study site, the neighbouring plots harboured similar fungal communities, even though the host species were different, and the effect of spatial distance on ECM fungal composition similarity was also suggested. Therefore, in order to clarify the preferred host species of individual ECM fungi in the field, further studies considering the spatial configuration of the host tree individual and spatial factors are necessary. The mechanisms by which host preference occurs, and further observation of the relationships between ECM fungal composition and host identities in other host species at wider environmental gradients, would be future research topics.

Data accessibility. All data of the 454 sequencing was shared in DRA (accession no. DRA007781, ftp://ftp.ddbj.nig.ac.jp/ddbj_database/dra/fastq/DRA007/DRA007781/. Sequence Read Archive (SRA) accession no. DRP003171)
Authors' contributions. S.M., T.O. and R.T. designed the study and S.M., S.I., T.O. and RT contributed to field survey and sampling. S.M., Y.S. and E.K. contributed to molecular experiments. S.M. and Y.S. analysed the data and interpreted the results. S.M., Y.S., R.T. and T.O. wrote the initial draft of the manuscript. All other authors critically reviewed the manuscript.
Competing interests. We declare we have no competing interests.
Funding. This study received partial financial support from Japan Society for the Promotion of Science (JSPS) to S.M. (17K15199) and T.O. (18K05731).
Acknowledgments. We thank the staffs of Hokkaido Forest Research Station, Field Science Education and Research Center, Kyoto University and Celina Sakaguchi for assistance in fieldwork; Miyuki Hirata for assistance in laboratory work; Koichi Ito for useful discussions and Hirotoshi Sato and Hideyuki Doi for critical comments on the manuscripts.

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
