## [Reviewer comments · Royal Society Open Science]

Review History

RSOS-191952.R0 (Original submission)

Review form: Reviewer 1 (Roger Koide)

Is the manuscript scientifically sound in its present form?

Yes

Are the interpretations and conclusions justified by the results?

Yes

Is the language acceptable?

No

Do you have any ethical concerns with this paper?

No

Have you any concerns about statistical analyses in this paper?

No

Recommendation?

Accept with minor revision (please list in comments)

Comments to the Author(s)

This is a valuable contribution as it perhaps the only one that clearly separates host effects from often confounding variables such as local environmental conditions and dispersion limitations.

There are numerous other places where the English needs to be corrected (see some examples below), but I will not attempt to correct the manuscript in every instance. The introduction is probably in most need of English correction. The other sections are not as bad. It should be a relatively simple matter to have an English-speaking scientist quickly go over the manuscript and correct the grammar.

Otherwise, the manuscript is scientifically sound. It arrives at logical conclusions that are of interest to mycorrhizal ecologists. I recommend only that the English be improved, particularly in the introduction.

Some suggestions for improved English:

Lines 35-36. "...38 ECM OTUs were only detected from particular host tree species."

Lines 42-43. This sentence should be rewritten. As it stands, it sounds as if ectomycorrhizal fungi only associate with members of these three plant families, which is not the case. It could be rewritten: "...Fagaceae, Betulaceae, and Pinaceae, among others, and represent...".

Line 51 "communities".

Line 53. "communities are simultaneously affected by each of these factors."

Line 54. "now try to quantify the effect of each factor separately."

Line 58. "...a variety"

Fig. 3. "Spatial variables" is misspelled.

There are many other such minor problems of English.

Review form: Reviewer 2

Is the manuscript scientifically sound in its present form?

Yes

Are the interpretations and conclusions justified by the results?

Yes

Is the language acceptable?

Yes

Do you have any ethical concerns with this paper?

No

Have you any concerns about statistical analyses in this paper?

No

Recommendation?

Accept with minor revision (please list in comments)

Comments to the Author(s)

Overall, this paper is well-constructed and of interest. It is not particularly novel, but advances knowledge of host preference and ecology of particular fungal taxa. Adding to and better integrating understanding of fungal biology would improve this paper.

28 How are neighboring conspecific plants distinguishable from host preference for that species? Also, there have been studies sequencing plant DNA from root tips to show preference for a host taxon vs. environment

33 Some indication of methods used in the abstract would be helpful

36 Unclear – were there 38 OTUs that were detected on only one of the host species?

43 This implies that ECM are only found on these three families, rephrase

52 change to “Practically, ECP fungal communities in the field are”.

68 This would be a good place to say something about the scale of variation in environmental conditions vs. scale of ECM fungi

74 Fungal mycelium is almost certainly more important than spore drop – studies show one or few fungal genetic individuals present at meter scales.

109 What is typical temp range and seasonality of rainfall?

116 *Alnus* are known to associate with a distinctive community of ECM fungi, this should be referenced

119 Curious whether lower diversity of ECM on newer stands due to lower species accumulation time

130 Report ranges of environmental variables in text – these sites are not very different and I’m not surprised you didn’t find differences

142 Did you select ECM-colonized tips? Did you try to select different ECM morphotypes?

153 These samples seem close together given the size of the plots, and a W-shaped sampling pattern is typically better at capturing variation. Is there soil type data for this area (e.g. percent sand, loam, clay?) and does it vary among plots?

194 Describe how trimming was performed.

196 Fix citation – can delete “software available online” since URL in cite

221 What R package? Cite is for base R, need package name and citation also

296 Just call the environmental fraction host tree species since that’s the only component, it’s confusing as written. Also, its contribution is reported as 28.9% in fig versus 28.7% here. Fig. 3 doesn’t convey additional information and could be omitted.

317 Rephrase for clarity, since you include host, a biotic factor, as an environmental factor in the analyses

320 Explained instead of “divided”

325 “were preferentially associated with” instead of “showed preference to” - this leaves open the possibility that hosts influence association

329 “clustered” isn’t a great word choice - “determined”?

329 did you test phylogeny distinct from host species?

337 If there’s selection of fungal partners by host plants, it is likely already occurring in the field. You don’t have the ability to distinguish between fungal preference and plant preference

358 Make this paranthetical its own sentence

383 better to say OTU is preferentially associated with *Betula* rather than attribute it to fungal control

384 “colonization” rather than “infection” throughout this paragraph

392 Most variation was unexplained, you don’t know what “primarily” structures ECM communities

Decision letter (RSOS-191952.R0)

08-Jan-2020

Dear Dr Matsuoka

On behalf of the Editors, I am pleased to inform you that your Manuscript RSOS-191952 entitled "Evaluation of host effects on ectomycorrhizal fungal community compositions in a forest landscape in northern Japan" has been accepted for publication in Royal Society Open Science subject to minor revision in accordance with the referee suggestions. Please find the referees' comments at the end of this email.

The reviewers and handling editors have recommended publication, but also suggest some minor revisions to your manuscript. Therefore, I invite you to respond to the comments and revise your manuscript.

- Ethics statement

- Data accessibility

It is a condition of publication that all supporting data are made available either as supplementary information or preferably in a suitable permanent repository. The data accessibility section should state where the article's supporting data can be accessed. This section should also include details, where possible of where to access other relevant research materials such as statistical tools, protocols, software etc can be accessed. If the data has been deposited in an external repository this section should list the database, accession number and link to the DOI

for all data from the article that has been made publicly available. Data sets that have been deposited in an external repository and have a DOI should also be appropriately cited in the manuscript and included in the reference list.

If you wish to submit your supporting data or code to Dryad (<http://datadryad.org/>), or modify your current submission to dryad, please use the following link:
<http://datadryad.org/submit?journalID=RSOS&manu=RSOS-191952>

- **Competing interests**

- **Authors' contributions**

- **Acknowledgements**

- **Funding statement**

Because the schedule for publication is very tight, it is a condition of publication that you submit the revised version of your manuscript before 17-Jan-2020. Please note that the revision deadline will expire at 00.00am on this date. If you do not think you will be able to meet this date please let me know immediately.

When submitting your revised manuscript, you will be able to respond to the comments made by the referees and upload a file "Response to Referees" in "Section 6 - File Upload". You can use this to document any changes you make to the original manuscript. In order to expedite the

processing of the revised manuscript, please be as specific as possible in your response to the referees. We strongly recommend uploading two versions of your revised manuscript:

If your manuscript is newly submitted and subsequently accepted for publication, you will be asked to pay the article processing charge, unless you request a waiver and this is approved by Royal Society Publishing. You can find out more about the charges at <https://royalsocietypublishing.org/rsos/charges>. Should you have any queries, please contact openscience@royalsociety.org.

on behalf of Dr James Locke (Associate Editor) and Kevin Padian (Subject Editor)

Associate Editor Comments to Author (Dr James Locke):

Please follow the clear directions provided by the reviewers to improve the manuscript for publication. As mentioned by the reviewers, please pay special attention to the grammar and spelling.

Reviewer comments to Author:

Reviewer: 1

Comments to the Author(s)

This is a valuable contribution as it perhaps the only one that clearly separates host effects from often confounding variables such as local environmental conditions and dispersion limitations.

There are numerous other places where the English needs to be corrected (see some examples below), but I will not attempt to correct the manuscript in every instance. The introduction is probably in most need of English correction. The other sections are not as bad. It should be a relatively simple matter to have an English-speaking scientist quickly go over the manuscript and correct the grammar.

Otherwise, the manuscript is scientifically sound. It arrives at logical conclusions that are of interest to mycorrhizal ecologists. I recommend only that the English be improved, particularly in the introduction.

Some suggestions for improved English:

Lines 35-36. "...38 ECM OTUs were only detected from particular host tree species."

Lines 42-43. This sentence should be rewritten. As it stands, it sounds as if ectomycorrhizal fungi only associate with members of these three plant families, which is not the case. It could be rewritten: "...Fagaceae, Betulaceae, and Pinaceae, among others, and represent...".

Line 51 "communities".

Line 53. "communities are simultaneously affected by each of these factors."

Line 54. "now try to quantify the effect of each factor separately."

Line 58. "...a variety"

Fig. 3. "Spatial variables" is misspelled.

There are many other such minor problems of English.

Reviewer: 2

Comments to the Author(s)

Overall, this paper is well-constructed and of interest. It is not particularly novel, but advances knowledge of host preference and ecology of particular fungal taxa. Adding to and better integrating understanding of fungal biology would improve this paper.

28 How are neighboring conspecific plants distinguishable from host preference for that species?

Also, there have been studies sequencing plant DNA from root tips to show preference for a host taxon vs. environment

33 Some indication of methods used in the abstract would be helpful

36 Unclear – were there 38 OTUs that were detected on only one of the host species?

43 This implies that ECM are only found on these three families, rephrase

52 change to “Practically, ECP fungal communities in the field are”.

68 This would be a good place to say something about the scale of variation in environmental conditions vs. scale of ECM fungi

74 Fungal mycelium is almost certainly more important than spore drop – studies show one or few fungal genetic individuals present at meter scales.

109 What is typical temp range and seasonality of rainfall?

116 *Alnus* are known to associate with a distinctive community of ECM fungi, this should be referenced

119 Curious whether lower diversity of ECM on newer stands due to lower species accumulation time

130 Report ranges of environmental variables in text – these sites are not very different and I’m not surprised you didn’t find differences

142 Did you select ECM-colonized tips? Did you try to select different ECM morphotypes?

153 These samples seem close together given the size of the plots, and a W-shaped sampling pattern is typically better at capturing variation. Is there soil type data for this area (e.g. percent sand, loam, clay?) and does it vary among plots?

194 Describe how trimming was performed.

196 Fix citation – can delete “software available online” since URL in cite

221 What R package? Cite is for base R, need package name and citation also

296 Just call the environmental fraction host tree species since that’s the only component, it’s confusing as written. Also, its contribution is reported as 28.9% in fig versus 28.7% here. Fig. 3 doesn’t convey additional information and could be omitted.

317 Rephrase for clarity, since you include host, a biotic factor, as an environmental factor in the analyses

320 Explained instead of “divided”

325 “were preferentially associated with” instead of “showed preference to” - this leaves open the possibility that hosts influence association

329 “clustered” isn’t a great word choice - “determined”?

329 did you test phylogeny distinct from host species?

337 If there's selection of fungal partners by host plants, it is likely already occurring in the field. You don't have the ability to distinguish between fungal preference and plant preference

358 Make this paranthetical its own sentence

383 better to say OTU is preferentially associated with *Betula* rather than attribute it to fungal control

384 "colonization" rather than "infection" throughout this paragraph

392 Most variation was unexplained, you don't know what "primarily" structures ECM communities

Author's Response to Decision Letter for (RSOS-191952.R0)

See Appendix A.

Decision letter (RSOS-191952.R1)

27-Jan-2020

Dear Dr Matsuoka,

It is a pleasure to accept your manuscript entitled "Evaluation of host effects on ectomycorrhizal fungal community compositions in a forested landscape in northern Japan" in its current form for publication in Royal Society Open Science. The comments of the reviewer(s) who reviewed your manuscript are included at the foot of this letter.

on behalf of Dr James Locke (Associate Editor) and Kevin Padian (Subject Editor)
openscience@royalsociety.org

Associate Editor Comments to Author (Dr James Locke):

This thorough revision has addressed the concerns of the reviewers and is now suitable for publication.

Appendix A

21-Jan-2020

RSOS-191952

"Evaluation of host effects on ectomycorrhizal fungal community compositions in a forest landscape in northern Japan"

Associate Editor Comments to Author:

Please follow the clear directions provided by the reviewers to improve the manuscript for publication. As mentioned by the reviewers, please pay special attention to the grammar and spelling.

(Response)

Thank you for comments to the manuscript. I have revised the manuscript thoroughly according to the comments by Handling Editor and Reviewers #1 and #2. The manuscript is again checked by English-language editing service. The modification points and detailed answers to the comments are listed below. Thank you again for considering the revised manuscript.

Reviewer comments to Author:

Reviewer: 1

Comments to the Author(s)

This is a valuable contribution as it perhaps the only one that clearly separates host effects from often confounding variables such as local environmental conditions and dispersion limitations.

There are numerous other places where the English needs to be corrected (see some examples below), but I will not attempt to correct the manuscript in every instance. The introduction is probably in most need of English correction. The other sections are not as bad. It should be a relatively simple matter to have an English-speaking scientist quickly go over the manuscript and correct the grammar.

Otherwise, the manuscript is scientifically sound. It arrives at logical conclusions that are of interest to mycorrhizal ecologists. I recommend only that the English be improved, particularly in the introduction.

(Response)

Thank you for critical reading and positive comments to the manuscript. I have revised the manuscript thoroughly according to your comments. The manuscript is again checked by English-language editing service.

Some suggestions for improved English:

Lines 35-36. "...38 ECM OTUs were only detected from particular host tree species."

(Response)

We revised as your suggestion. (L35-36)

Lines 42-43. This sentence should be rewritten. As it stands, it sounds as if ectomycorrhizal fungi only associate with members of these three plant families, which is not the case. It could be rewritten: "...Fagaceae, Betulaceae, and Pinaceae, among others, and represent...".

(Response)

Thank you for the comment. We revised as your suggestion. (L42-43)

Line 51 "communities".

(Response)

We revised as your suggestion. (L51)

Line 53. "communities are simultaneously affected by each of these factors."

(Response)

We revised as your suggestion. (L53)

Line 54. "now try to quantify the effect of each factor separately."

(Response)

We revised as your suggestion. (L54)

Line 58. "...a variety"

(Response)

We revised as your suggestion. (L57)

Fig. 3. "Spatial variables" is misspelled.

(Response)

Thank you for the comment. We removed Fig. 3 according to the comment from reviewer

2.

There are many other such minor problems of English.

(Response)

The manuscript is again checked by English-language editing service.

Reviewer: 2

Comments to the Author(s)

Overall, this paper is well-constructed and of interest. It is not particularly novel, but advances knowledge of host preference and ecology of particular fungal taxa. Adding to and better integrating understanding of fungal biology would improve this paper.

(Response)

Thank you for critical reading and useful comments to the manuscript. I have revised the manuscript thoroughly according to your comments. The modification points and detailed answers to the comments are listed below.

28 How are neighboring conspecific plants distinguishable from host preference for that species? Also, there have been studies sequencing plant DNA from root tips to show preference for a host taxon vs. environment

(Response)

Thank you for the comments. In this time, the effects of neighboring conspecific plants cannot be distinguished, and we don't have ideas for distinguishing them. Therefore, we removed "conspecific" from the sentence. (L28)

In addition, explanation and citation of studies sequencing plant DNA to show preference are incorporated into the Introduction section. (L62-64)

33 Some indication of methods used in the abstract would be helpful

(Response)

Indication of methods are incorporated into the abstract (L32-33). Thank you.

36 Unclear – were there 38 OTUs that were detected on only one of the host species?

(Response)

Yes. We revised the sentence. "38 ECM OTUs were only detected from particular host

tree species.” (L36)

43 This implies that ECM are only found on these three families, rephrase

(Response)

We revised the sentence. “Ectomycorrhizal (ECM) fungi are symbionts of tree species belonging to the families Fagaceae, Betulaceae, and Pinaceae, among others, and represent a dominant group of microorganisms inhabiting temperate and boreal forest floors” (L43)

52 change to “Practically, ECP fungal communities in the field are”.

(Response)

We revised the sentence according to the comment from reviewer 1 and removed “in field” from the sentence (L52-53).

68 This would be a good place to say something about the scale of variation in environmental conditions vs. scale of ECM fungi

(Response)

We fully agree with the comment on the importance of scale. So far, however, we don't know exactly what environmental or spatial scale would be wide or narrow for ECM (Bogar & Peay 2017). The scale dependence is an important issue to be evaluated in the future. Therefore, we added citations and discussion on the potential effect of scale on the results in the discussion section rather than in the introduction section (L365-366, L413-414).

74 Fungal mycelium is almost certainly more important than spore drop – studies show one or few fungal genetic individuals present at meter scales.

(Response)

Thank you for the comment. Considering the importance of hyphae, the order of sentences was changed. (L75-78)

109 What is typical temp range and seasonality of rainfall?

(Response)

The 30-year mean monthly temperature is highest in August (19.8 °C) and lowest in January (− 9.0 °C), and the 30-year mean monthly precipitation is highest in September (181.8 mm) and lowest in January (30.7 mm). These climatic data are updated and shown in the Materials and Methods section. (L111-115)

116 *Alnus* are known to associate with a distinctive community of ECM fungi, this should be referenced

(Response)

The association of *Alnus* with a distinctive ECM community is mentioned in the text. (L124-125)

119 Curious whether lower diversity of ECM on newer stands due to lower species accumulation time

(Response)

The relationship between ECM richness and forest stand age is a theme of our future research. However, no significant relationship has been found between the two in this study sites.

130 Report ranges of environmental variables in text – these sites are not very different and I'm not surprised you didn't find differences

(Response)

Ranges of environmental variables are shown in the Materials and Methods section. (L138-139, L168-169)

142 Did you select ECM-colonized tips? Did you try to select different ECM morphotypes?

(Response)

We selected ECM-colonized tips but did not try to select different ECM morphology. This explanation is incorporated into the Materials and Methods section. (L150-151)

153 These samples seem close together given the size of the plots, and a W-shaped sampling pattern is typically better at capturing variation. Is there soil type data for this area (e.g. percent sand, loam, clay?) and does it vary among plots?

(Response)

Thank you for the comment. Our sampling method has some improvements. Thus the potential effect of the sampling on the result is discussed in the discussion section (L366). This time, however, variations in ECM composition among host tree species have been successfully detected. Your advice on sampling methods will be used for future research. Soils of the study sites are characterized as Andosols (IUSS Working Group WRB, 2015) and soil texture is characterized as clay loam or loam texture. These soil type and texture

do not vary among plots. The explanation of soils are incorporated into the Materials and Methods section. (L116-117)

194 Describe how trimming was performed.

(Response)

The pyrosequencing reads were trimmed with a minimum quality value of 27 at the 3' tails. The description is incorporated into the sentence. (L203)

196 Fix citation – can delete “software available online” since URL in cite

(Response)

We remove “software available online” from the sentence. (L205)

221 What R package? Cite is for base R, need package name and citation also

(Response)

“Package” in this sentence was wrong. We remove “package” from the sentence (L230). The functions and packages used in the individual analyses are shown in the text.

296 Just call the environmental fraction host tree species since that's the only component, it's confusing as written. Also, its contribution is reported as 28.9% in fig versus 28.7% here. Fig. 3 doesn't convey additional information and could be omitted.

(Response)

We replace “host tree species” with “environmental fraction” (L304-306). Previous figure 3 is removed from the manuscript because there was no additional information.

317 Rephrase for clarity, since you include host, a biotic factor, as an environmental factor in the analyses

(Response)

We revised the sentence. “We quantitatively evaluated the effect of host tree species, abiotic environments and spatial factors on ECM fungal communities in the field...” (L325-327).

320 Explained instead of “divided”

(Response)

We replace “divided” with “explained”. (L328)

325 “were preferentially associated with” instead of “showed preference to” - this leaves

open the possibility that hosts influence association

(Response)

We replace “showed preference to” with “were preferentially associated with”. (L333)

329 “clustered” isn’t a great word choice - “determined”?

(Response)

Thank you for the comment. We replace “clustered” with “determined”. (L337)

329 did you test phylogeny distinct from host species?

(Response)

No. We removed “phylogeny” from the sentence. (L337)

337 If there’s selection of fungal partners by host plants, it is likely already occurring in the field. You don’t have the ability to distinguish between fungal preference and plant preference

(Response)

Thank you for the critical comment. We agree that we don’t have the ability of distinguish between fungal preference and plant preference. This notification is incorporated into the discussion section. (L342-343)

358 Make this paranthetical its own sentence

(Response)

We revised the sentence as your suggestion. (L369)

383 better to say OTU is preferentially associated with Betula rather than attribute it to fungal control

(Response)

Thank you for the comment. We replaced “This OTU is preferentially associated with Betula” with “This OTU prefers Betula” (L394).

384 “colonization” rather than “infection” throughout this paragraph

(Response)

We replaced “infection” with “colonization” throughout this paragraph. (L396-398)

392 Most variation was unexplained, you don’t know what “primarily” structures ECM communities

(Response)

Agree. We removed “primarily” from this sentence. (L404)